# Coping, health anxiety, and stress among health professionals during Covid-19, Cape Coast, Ghana

Anthony K. Nkyi[1]*, Bridgette Baaba[2]

**1** Department of Guidance and Counselling, College of Education Studies, University of Cape Coast, Cape Coast, Ghana, **2** Counselling Center, College of Education Studies, University of Cape Coast, Cape Coast, Ghana

* ankyi@ucc.edu.gh

## Abstract

The aim of this study is to investigate coping strategies, health anxiety and stress among healthcare professionals in selected hospitals in the Cape Coast Metropolis during the Covid 19 pandemic. This study adopted the Descriptive survey design. The Multistage sampling technique was used to select 322 health professionals. The health professionals included Medical Officers, Physician Assistants and Nurses. Data were gathered using the Health Anxiety Inventory (HAI-SF), Perceived Stress Scale (PSS), and the Brief-COPE Inventory. Analyses were conducted using mean and standard deviation, ANOVA as well as Independent Samples t-test. Results indicate that Health professionals adopted diverse coping strategies ranging from positive to negative coping style to overcome the negative impact of the COVID-19 pandemic. Results also indicate that coping strategies significantly influenced health anxiety levels of health professionals, and that Active Coping is a significant determinant of stress among health professionals. Male health professionals had significantly more health anxiety than females. However, gender was not a significant factor in the experiences of stress. Lastly, age of health professionals does not determine the type of coping strategy they adopted during the pandemic.

## Introduction

COVID-19, an infectious disease which is linked with a higher prevalence of morbidity and mortality worldwide [1], has caused significant social, economic and health crises [2]. Health professionals are more highly susceptible to this infectious disease due to their essential role as health providers [3] The growing number of discovered cases among the general population and health workers, lack of adequate protective equipment, lack of medication and support, are significant predictors of occupational burnout among health workers [4].

Many challenges were related to significant levels of psychological distresses [5–7], during the outbreak of the SARS (between 2002 and 2004). Many health professionals considered resignation because they were extremely stigmatized and feared spreading and infecting their family and friends.

**Data Availability Statement:** All relevant data are within the paper and its Supporting Information files.

**Funding:** The authors received no specific funding for this work. The funders had no role in study design, data collection and analysis, decision to publish, or preparation of the manuscript.

**Competing interests:** The authors have declared that no competing interests exist.

A study by Ofori *et al.* [8] found that among health workers in Ghana, the coping strategies adopted to mitigate the psychological impact of the COVID-19 pandemic included praying more often. Other factors that aided in the reduction of the effect of the pandemic include positive attitude from colleagues, and the government's tax-free relief salary. Asare-Nuamah, Onumah, Dick-Sagoe, and Kessie [9] studied the perception and coping strategies for COVID-19 among rural Ghana indigenes. The qualitative descriptive phenomenological approach was adopted. The results indicated that many adopted the use of traditional methods, such as drinking locally prepared herbs, as well as following the COVID-19 preventive protocols, to cope with the effect of the pandemic.

Oti-Boadi [10] studied the effect of the pandemic on the psychological well-being of university students in Ghana. The qualitative descriptive phenomenological approach was adopted. The results indicated that coping strategies, such as denial, venting, and humour were adopted. The analysis of the study showed that only maladaptive coping was found to be significantly and positively associated with fear of COVID-19. Again, both adaptive coping and maladaptive coping strategies had a mediating effect on fear of COVID-19 and psychological distress.

Savitsky *et al.* [11] evaluated the prevalence of anxiety during this COVID-19 era and coping methods employed among nursing students in the Southern District of Israel. The descriptive study engaged 244 nursing students. Findings revealed moderate (42.3%) and severe (13.1%) levels of anxiety among participants. The high level of anxiety was closely related to a lack of protective equipment, fear of infection, and gender. Coping methods included adopting humour and building strong resilience.

Riaz *et al.* [12] through a descriptive survey, assessed the mental health outcomes and coping strategies of frontline health care workers and hospital supporting personnel, like administrative workers and laboratory personnel. The researchers, through an online questionnaire, sought data for the project. One of the major aims of the study was to identify the causes of stress and the major coping methods employed by these personnel in the face of the pandemic. In all, a total of one hundred and twenty-seven (127) frontline medical workers completed the online questionnaire. The findings from the study proved that health care workers during the face of the COVID-19 pandemic faced both emotional and psychological stress. Moreover, the researchers identified that the major coping strategy employed by medical practitioners in combating the stress was emotion-focused coping technique, as enshrined in Lazarus and Folkman's Transactional Theory of Stress and Coping. The findings from this study are crucial for the current study because it espouses some major coping technique employed by healthcare workers in their quest to cater for patients within the pandemic.

Also, Munawar & Choudhry [13] adopted a framework thematic analysis approach to examine the stress-coping approach adopted by frontline emergency health workers dealing with Covid-19 in Pakistan. Data was collected through three channels: face-to-face interview, semi-structured interviews, and interviews through telephone. The findings from the study emphasized that coping mechanisms such as checking media exposure and low sharing of COVID-19 details, and religious coping, among others, were some of the coping mechanisms adopted by the health care workers in Pakistan to help combat the stress of the COVID-19 pandemic. The findings from this study, though it was undertaken in Pakistan, provide important coping strategies that health care workers used during the period of the pandemic, and this is crucial for the study as the study seeks to examine the coping strategies employed by primary health care workers.

The reviewed studies showed shreds of evidence of association of psychological illness with COVID-19 disease. Particularly, moderate to high levels of anxiety, depression, and stress were found to be linked with the coronavirus disease. Factors such as excessive modern media use, feeding on wrong sources of information, and socio-economic factors (educational level, age,

marital status, cohabiting with family, gender, and being unemployed by the government) also predicted high levels of generalized anxiety. Lack of COVID-19 protective equipment and fear of infection also predicted psychological disorders. Some psychological impacts of COVID-19 on health professionals, namely, anxiety, depression, stress, and insomnia were also identified by some studies. The studies also established that coping practices, namely adopting humour, building strong resilience, adherence to diet and health practices, appropriate self-distancing from coronavirus-related news, involvement in hobbies, and staying indoors, serve as buffers to mental illness.

## Health anxiety

The discovery and spread of the contagious COVID-19 have taken a significant strain on the health system and health professionals. Beside interests for their welfare, health care personnel fear infecting their closest relatives. Whereas some level of fear is vital for our survival, excessive fear has harmful effects and can end in mental disorders such as phobia and social anxiety [14] (Mertsens, Gerritsen, Dunidam, Salemink, and Engelhard [15] opined that the level of anxiety among the people has substantially increased as compared to studies conducted before the onset of the pandemic. Tyrer [16]) asserted that as the pandemic dominates every news and social media platform, health anxiety has become necessary to consider.

Health anxiety is also known as illness anxiety and was formerly called hypochondria. Tyrer [16] indicated that health anxiety is similar to hypochondriasis but is characterized by the fear of illness instead of the conviction that one has acquired an illness. According to Hart and Björgvinsson (2010), the difficulty in the diagnosis of hypochondriasis led to recent use of the term *health anxiety*. Abramowitz, Deacon, and Valentiner [17] described health anxiety as "the tendency to misinterpret normal or benign physical symptoms and believe that one has or is acquiring a serious illness, in the absence of any actual illness." Health anxiety is also similar to a novel diagnosis called illness anxiety disorder, according to the fifth edition of the Diagnostic and Statistical Manual of Mental Disorders, also known as DSM-5. [18] Some risk factors include a time of major life stress, the threat of serious illness such as epidemic or pandemic, history of abuse as a child, and personality traits [19] (Te Poel, Baumgartner, Hartmann, & Tanis, 2016).

## Stress

Stress is a multidimensional concept and occurs when our physical and psychological strength cannot handle threats from the environment [20]. According to Selye [21] Stress refers to "the bodily processes that result from circumstances that place physical or psychological demands on an individual." In its basic terms, stress refers to the body's reaction to change that needs response. The body, in the face of any adjustment, reacts with emotional, physical, and mental responses. Ismail *et al.* [22] established that stress may be divided into positive (good stress or eustress) and negative stress (distress). Whiles eustress aid in higher performance of activity, distress decreases the motivation to engage in quality work and is associated with health problems [23]. According to Seiler *et al.* [24] over the past three decades, studies have proved beyond doubt that stress has a huge impact on clinically important immune system outcomes. These, according to the scholar, include inflammatory processes, vaccination, and wound healing, among others. It is evident that stress is needed to the point where it helps boost one's performance; however, excessive stress is harmful. Lastly, it is imperative to note that stress is a state and should not be classified as illness [25].

A bird's eye view into existing literature will reveal the numerous types of stress pointed out by scholars. For instance, according to Bhowmik, Vel, Rajalakshmi, and Kumar [26] there are

three kinds of stress. They are routine stress, stress caused by variation and traumatic stress. According to the researcher, routine stresses are the types of stress, which are usually connected to the daily pressures, and demands of work. These types of stress are brought about through our daily responsibilities. The second type of stress, according to Bhowmik et al. [26], is the type that occurs due to a sudden negative change in the life of an individual. Examples of such negative changes includes losing one's job, illness, sickness, or divorce. The last type of stress revealed by the researchers is traumatic stress. According to them, traumatic stress occurs when an individual is in danger of being hurt or killed. War, fatal accidents, natural disasters, and assaults are some of the events that lead to traumatic stress. Traumatic stress tends to cause Post-Traumatic Stress Disorder (PTSD).

On the other hand, Agarwal and Malhotra [27], in their quest to research into the stress within the workplace, pointed out three forms of stress, namely, Acute Stress, Episodic Acute Stress and Chronic Stress. These kinds of stress provide a clear understanding of the specific nature of stress that healthcare providers can encounter because of the COVID-19 pandemic. Thus, during this pandemic, healthcare providers' experiences of stress may range from an acute stress which lasts briefly, to chronic stress which demands continuous psychological intervention and plan.

The foregoing makes it clear that stress may create negative consequences in the life of individuals. Likewise, stress among health professionals may lead to negative consequences on their lives and may extend to the quality of care they provide for patients [28] Earlier research, through past epidemics (SARS and MERS), shows that health care personnel recounted significant stress levels [29,30] The effect of this crisis on health personnel might increase their stress and this could be connected with other disorders like anxiety and depression [31]. With an increasing demand for healthcare, health professionals are required to work long hours and are deprived of the required resources to work effectively.

## Coping

Coping is a usual dominant extenuating feature in models of health, fear, and pain. Salkovskis, Rimes, Warwick, & Clark, Man, and Lazarus [32] revealed that coping strategies are the "behavioural and cognitive efforts that help to reduce the pressure of stressful events and are available (mental) resources to mitigate potential threat." Coping strategies are the behaviours and thoughts used to regulate the exterior and inner demands of situations that are regarded as stressful. Folkman and Lazarus [33] defined coping as the behavioural and cognitive exertions made to accommodate or lessen the internal or outward pressures and conflicts. According to these researchers, there are two ways of coping, namely, Emotional Coping strategy and Problem-focused strategy. Coping is a basic process fundamental to survival and adaptation. Coping helps individuals identify, assess, deal with, and learn from stressful situations. According to Townsend & Wells [34] coping is the ability to control challenging, threatening and potentially harmful events, which are critical to one's wellbeing. Coping strategies can be behavioural or cognitive. Cognitive coping strategies encompass the conscious control of one's emotions and thoughts. On the other hand, behavioural coping strategies are verbal and physical activities used to control stress.

Folkman and Moskowitz [35] also revealed four kinds of coping strategies, i.e.: Positive appraisal, problem-focused strategy, emotion-focused approach and meaning-focused strategy. A positive appraisal is a type of coping strategy where an individual reframes a situation and tries only to see the positive side of the event at hand. This type of coping strategy yields a positive effect on the individuals' health and entire wellbeing. The problem-focused approach is the type of coping strategy in which an individual directs all efforts to solve a stressor at

hand or finding solutions to a particular problem causing distress. The problem-focused coping approach helps gather resources to solve a problem. It is sometimes known as the task-oriented coping strategy. An emotion-focused coping strategy is that type of coping mechanism where individuals take the step to manage emotional distress. It includes cognitive behaviours such, as looking at the positive side, and it also has behavioural strategies, such as using drugs or seeking emotional support. Lastly, meaning-focused coping strategy, as espoused by Folkman and Moskowitz, involves looking for meanings into troubles or adversities.

Wasim, Raana, Bushra and Riaz [2] revealed that though physical wellbeing of health professionals (provision of personal protective equipment, adequate training to prevent infections, and other safety measures) have been taken care of by the health sectors in many countries, the mental health needs of health professionals have received little or less attention in many countries. Wasim and colleagues found that health professionals experienced a severe level of anxiety, insomnia, stress, and depression which could lead to attention deficit, impairment in cognitive function and clinical decision-making [2] The presence of such impairment will mostly affect the performance of health professionals. Chang, Xu, Rebaza, Sharma, and Cruz [36] established that for optimum delivery of quality health care to the populace, the wellbeing and mental health of healthcare professionals are important to consider. Without adequate measures and interventions, the impact of coronavirus will have debilitating and enduring consequences on health professionals. It appears that empirical studies assessing the psychological impact of COVID-19 on health workers are limited in Ghana, especially in the Central Region. While few studies have been conducted on the psychological impact of COVID-19 pandemic in Ghana, even fewer studies still have been dedicated to the psychological wellbeing of healthcare professionals who serve as the frontline workers [8,10,37,38]. The coping strategies adopted by healthcare professionals in some healthcare facilities in the Cape Coast Metropolis can help in the development of appropriate interventions and treatment for health professionals to improve their psychological well-being.

It is against this sordid background that this study investigated health anxiety and stress and ways of coping among health professionals during the current Covid-19 pandemic in Ghana.

## Research objectives

The study was to determine the coping strategies among healthcare professionals with anxiety and stress amid COVID-19. Specifically, the study addressed the following objectives:

1. Identify the coping strategies adopted by healthcare professionals to mitigate the effects of COVID-19 on their Health Anxiety and Stress levels in selected hospitals in the Cape Coast Metropolis.

2. Determine coping strategies' influence on health anxiety of health professionals.

3. Determine coping strategies influences on stress of health professionals.

4. Identify the difference between male and female healthcare professionals in terms of Health Anxiety and Stress among healthcare professionals in selected hospitals in the Cape Coast Metropolis.

5. Ascertain the difference among age categories in terms of Coping Strategies among healthcare professionals in selected hospitals in the Cape Coast Metropolis.

## Research question

1. What coping strategies are used by healthcare professionals to mitigate the effects of health anxiety and stress in their COVID-19-related responsibilities in the Cape Coast Metropolis?

## Hypotheses

1. $H_a$: There is statistically significant influence of coping strategies on health anxiety of health professionals.

2. $H_a$: There is statistically significant influence of coping strategies on stress level of health-care professionals in the Cape Coast Metropolis.

3. $H_A$: There is a significant difference among male and female health professionals in terms of health anxiety and stress, in selected hospitals in the Cape Coast Metropolis.

4. $H_A$: There is a significant difference among age category of health professionals in terms of coping strategies in selected hospitals in the Cape Coast Metropolis.

## Research design

This study adopted the quantitative research methodology, where the aim is to numerically quantify the collection and analysis of the data (Bryman, 2012). Specifically, the descriptive survey was used for the study.

## Study area

The study was conducted in the Cape Coast Teaching Hospital (CCTH) and the University of Cape Coast Hospital (UCC-H) in the Cape Coast Metropolis, the capital town of the Central Region of Ghana. These hospitals are used as Coronavirus case centres.

## Population

The target population was made up of registered and recognized healthcare professionals (Medical Officers, Physician Assistants, and Nurses) in the CCTH and the UCC-Hospital. These facilities served as COVID-19 emergency response units with special wards designated for confirmed and suspected cases on COVID-19. The estimated number of Medical Officers, Physicians Assistants, and Nurses in CCTH is 1,188 (Medical Officers 281, Physicians Assistants 4, and Nurses 903), while the UCC Hospital has a total number of 139 (Medical Officers 12, Physicians Assistants 7, and Nurses 120). The estimated population for the study from the two facilities is 1,327 health professionals. This is presented in Table 1.

## Inclusion and exclusion criteria

Participants were registered and licensed Medical Officers, Physician Assistants and Nurses working at the CCTH and UCC Hospitals. All participants were individuals who were at post

**Table 1. Distribution of estimated sample of health professionals.**

| Category of HP | Population | | Combined Population | Estimation of Health Professionals for the study | | |
|---|---|---|---|---|---|---|
| | | | | Proportionate computation sample | Estimate Sample | |
| | CCTH | UCC-H | | | CCTH (89%) | UCC-H (7%) |
| Medical Officers | 281 | 12 | 293 | (293 ÷ 1327) × 306 = **67** | 60 | 7 |
| Physicians Assistants | 4 | 7 | 11 | (11 ÷ 1327) × 306 = **3** | 2 | 1 |
| Nurses | 903 | 120 | 1023 | (1023 ÷ 1327) × 306 = **236** | 210 | 26 |
| **TOTAL** | **1,188** | **139** | **1,327** | **306** | **267** | **25** |

or on duty during the start of the Coronavirus pandemic and had not proceeded on leave or vacation. Health professionals who were on leave or vacation during the onset of the pandemic in Ghana were exempted from the study.

## Sampling procedure

The use of sampling involves a strategic decision by the researchers to focus on some, rather than all, of the research population [39] In other words, a sample of a study signifies the smaller group of people or cases selected to represent the entire population in a study. On the other hand, sampling denotes the procedures involved in choosing representatives out of the population for a study [40,41]. The sample size was 306 health professionals in the selected hospitals. This sample was determined using Gill, Johnson, and Clark [42]) sampling size determination table. The assumptions of this sample size determination table include a 95% confidence level, 5% margin of error, and a 50% variance of the population.

The sample size for the study was selected using the Multistage Sampling technique. First, the stratified sampling procedure was used. Under the stratified sampling technique, the entire population was divided into subgroups (strata) (Doctors, Physicians Assistants, and Nurses). A percentage was calculated to ensure that each of the group is fairly represented from each facility. The procedure is presented in Table 1.

The purposive sampling procedure was also adopted to choose eligible candidates for the research. By this sampling procedure, participants were chosen because they are health professionals from the CCTH and UCC Hospital. Additionally, participants were health professionals who were on duty during the onset of the pandemic in Ghana and had not taken a leave or proceeded on vacation.

After ensuring that each subgroup is fairly represented in terms of percentages, and eligibility, the convenience sampling procedure was used to select participants. The approach, under this method, considered participants who were readily available and willing to engage in the study. For this reason, data were gathered from available participants.

## Instruments

The data collection instruments consist of a demographic characteristics questionnaire, Brief-COPE Inventory, the Health Anxiety Inventory (HAI).and the perceived stress scale.

The demographic characteristics questionnaire included age, gender, job category and work experience,

Brief-COPE Inventory: Health professionals' style of coping with psychological burden related to COVID-19 was measured using the Brief-COPE, developed by Charles S. Carver in 1997 [43]. The scale was designed to assess proficient and unproductive ways of coping with adverse events. It has 28 items ranked on a 4-point Likert scale type, namely, 1 = "I haven't been doing this at all," 2 = "A little bit," 3 = "A medium amount" and 4 = "I've been doing this a lot." There are 14 subscales (each contains 2 items) which are "Self-distraction, Active coping, Denial, Substance use, Use of emotional support, Use of instrumental support, Behavioural disengagement, Venting, Positive reframing, Planning, Humour, Acceptance, Religion, & Self-blame." The 14 subscales measure coping style in two major ways, namely, "Approach Coping, and Avoidant Coping." The subscales of the Brief-COPE inventory showed reliable internal consistency between 0.50–0.73. According to Bai, Liu, Bo and Zhang [44] this measure demonstrated a reliable internal consistency of 0.84, and the Cronbach's alpha for the subscales ranged between 0.51–0.90.

The HAI-SF was adopted to examine experiences of health anxiety among health professionals. It is a self-report scale developed by Salkovskis, Rimes, Warwick and Clark [45] to

measure indications of illness related to anxiety and hypochondria. It includes 18 items, scored on a 4-point Likert scale ranging from 0 to 3 in the 3 components of worry about health (7 items; 0–21 points), awareness of bodily sensations or changes (6 items; 0–18 points), and feared consequences of having an illness (5 items; 0–15 points). The total score range of this questionnaire is 0–54 points. Scores of 0–18, 18–36, and above 36 indicate a low, moderate, and high health anxiety level respectively. Rabiei, Klantari, Asgari and Bahrami [46] revealed that the scale measures the extent to which people are worried about disease infection and the behaviours they would portray if they were to be infected with that disease. In other words, HAI-SF assesses an individual's anxiety related to perceived illness or their exact reactions if they were to be diagnosed with a serious health condition. The short form of this questionnaire was first developed by Salkoskis and Warwick [45] and included 18 items scored on a 4-point Likert scale ranging from 0 to 3, in the 3 components of worry about health (7 items; 0–21 points), awareness of bodily sensations or changes (6 items; 0–18 points), and feared consequences of having an illness (5 items; 0–15 points). The total score range of this questionnaire is 0–54 points. Scores of 0–18, 18–36, and above 36 indicate a low, moderate, and high health anxiety level respectively. A test value or hypothesized mean of 2.5 was determined as the standard against which the mean of means and the item mean would be compared. An obtained mean lesser than the test value shows participants experienced significantly less. health anxiety, whiles a score above the hypothesized mean reveals the experiences of significantly high levels of health anxiety. According to Salkovskis et al [45]., the scale has a good reliability coefficient (Cronbach Alpha = 0.89). A test-retest of the scale proved good reliability coefficient (r = 0.90).

The Perceived Stress Scale (PSS) was developed by Cohen and Williamson [47] Arguably, this scale is the most common scale for the measurement of stress among a varied population. The PSS used in this study measured the level of stress among health professionals during the COVID-19 pandemic in the Cape Coast Metropolis. Participants had to report on their feeling of being upset, ability to control irritations, the feeling of nervousness, confidence to handle personal issues, loss of control, and how often they became angry because things were out of control. The scale has 10-item scale rated on a 5-Likert scale ranging from 0 = Never to 5 = Very often. Participants were required to respond to the scale by indicating their agreement or disagreement with each item.

Individual scores on the PSS can range from 0 to 40, with higher scores indicating higher perceived stress. Scores ranging from 0–13 would be considered low stress. Scores ranging from 14–26 would be considered moderate stress. Scores ranging from 27–40 would be considered high perceived stress. A test value mean of 3.0 was determined as a criterion measure. A score above the test value indicates that participants experienced significantly higher levels of stress, whereas an obtained score lower than the test value shows participants experienced significantly lower stress. According to the authors, the PSS demonstrated good internal consistency reliability (Cronbach's alpha = 0.78).

Pilot Testing: The instruments were pilot tested at the Cape Coast Metropolitan Hospital. A total of 40 health professionals (5 Medical Officer, 6 Physician Assistants, and 29 nurses) were engaged in this exercise. Reliability Cronbach alpha of the instruments was health anxiety scale (18 items) 0.71 Perceived Stress Scale (PSS) 0.89. and Brief-COPE Inventory (28 items) 0.74.

## Ethical consideration

The study was approved by the Institutional Review Board (IRB) of the University of Cape Coast (UCC) before data collection took place. Upon approval, ethical clearance was acquired from the IRB and an introductory letter from the Department of Guidance and Counselling

was submitted to the Administrators of the CCTH and UCC-Hospital for permission to conduct the study.

Additionally, participants were duly educated on the details and requirements of the study, the importance of their involvement, as well as the voluntary nature of the study. Formal consent was sought from those who willingly decided to engage in the study by signing an informed consent form.

Similarly, anonymity and confidentiality of participants' information were strictly observed. In this regard, identity of participants was concealed. Neither names nor any identifiable information from participants was taken. Only the assigned index and numbers were used to identify the questionnaire during data entry. No aspect of information from participants was given out without their approval. The answered questionnaires were kept in a locked box and were only retrieved when needed for further entry or verification.

## Data collection procedures

Ethical clearance from the IRB of the University of Cape Coast (UCCIRB/CES/2021/39) and an Introductory Letter from the Department of Guidance and Counselling were submitted to the appropriate authorities of the CCTH and UCC–H for their approval to conduct a study in these health organizations. The ethical clearance spelt out the purpose of the study, the need for individual participation, anonymity as well as confidentiality of the participant's responses. After approval was given, all participants were informed about the aim, as well as the right to participate or disengage from this research. Eligible persons who agreed to partake in the study were educated on the requirement of the questionnaire and the proper ways to fill them. All participants signed an informed consent form. Due to the work schedule of health professionals, they were contacted at different time ranges. Approximately, data collection spanned two months.

## Data processing and analysis

All completed questionnaires were rechecked for consistency and completeness. Coding and computerization were done with SPSS version 27 after the creation of data analysis fields. The editing procedure helped check whether all items had been accurately responded to. Section A, which gathered data on demographic characteristics was analysed descriptively using frequencies and percentages. These included the participants' gender, age, and category of the health profession. Analyses were conducted using mean and standard deviation, Multiple Linear Regression Analysis ANOVA, as well as Independent Samples t-test.

## Results and discussions

### Results

Demographic data of participants. This sub-section presents and discusses the background characteristics of the participants, namely gender (male and female), age, and category of health professionals (Medical Doctors, Physician Assistants and Nurses). The result of the analysis is presented in Table 2.

The demographic information covers the gender, age, and category of health professionals. First, this section provided detailed information on the demographic (composition) features of health professionals sampled for the study. Then, research objective dealt with analysis involving the demographic characteristics of participants. These objectives sought to determine the differences in the category of participants regarding their demographic characteristics; thus, in terms of health anxiety and stress. This analysis provides readers another perspective from

**Table 2. Distribution of participants by demographic characteristics (n = 322).**

| Variable | Sub-scale | Frequency | Percentage% |
|---|---|---|---|
| **Gender** | Male | 80 | 24.8 |
| | Female | 242 | 75.2 |
| | **Total** | **322** | **100.0** |
| **Age (in years)** | 18–29 | 213 | 66.2 |
| | 30–49 | 107 | 33.2 |
| | 50–60 | 2 | 0.6 |
| | **Total** | **322** | **100.0** |
| **Category of Health Profession** | Medical Doctors | 19 | 5.9 |
| | Physician Assistants | 17 | 5.3 |
| | Nurses | 286 | 88.8 |
| | **Total** | **322** | **100.0** |

Source: Field survey, (2021).

which to judge experiences of health anxiety and stress among healthcare professional during the pandemic.

As shown in Table 2, the majority of participants, in terms of gender, were female health professionals (n = 242, 75.2%). Regarding age categories, most participants were between "*18–29 years*" (n = 213, 66.2%). The least represented group in terms of age category was "*50–60 years*" (n = 2, 0.6%). Lastly, nurses dominated the study, representing approximately two-thirds of the sample (n = 286, 88.8%). This result reflects the fact that nurses represent the highest percentage of health professionals in Ghana.

## Main results

**Coping strategies adopted by health professionals.** Objective one was to determine the significant coping strategies adopted by health professionals during the current COVID-19 pandemic. A 28-item scale was used to access the coping strategies used by participants. The scale was rated on a 4-point Likert type scale, namely, 1 (I haven't been doing this at all), 2 (A little bit), 3 (A medium amount), and 4 (I've been doing this a lot). There are 14 subscales. Mean and standard deviation was used to analyse the data. A test value of 2.5 was determined as a criterion measure. A score above the criterion or hypothesized mean shows participants frequently adopted what is commonly used by health professionals. A lower score, on the other hand, denotes that the coping strategy is less frequently adopted. A summary of the analyses is presented in Table 3.

The most significant coping strategy adopted by participants is *Active Coping (M = 5.39, SD = 1.74)*. This strategy was followed by *Planning (M = 5.30, SD = 1.59)* and *Religion (M = 5.30, SD = 1.68)*. Other significant responses include *Acceptance (M = 5.14, SD = 1.65)*. This strategy is followed by *Instrumental Support (M = 5.12, SD = 1.74)*. The least adopted coping strategy adopted by participants is *Substance Use (M = 3.47, SD = 1.63) and Behavioural Disengagement (M = 3.99, SD = 1.49)*.

## Coping strategies influence on health anxiety of health professional

Hypothesis one determines the extent to which coping strategies influence health anxiety of health professionals. Multiple regression analysis was conducted, and the results are presented in Table 4.

**Table 3. Analysis of results of coping strategies among health professionals.**

| Items | Mean | SD |
|---|---|---|
| 1. Active coping | 5.39 | 1.74 |
| 2. Planning | 5.30 | 1.59 |
| 3. Religion | 5.30 | 1.68 |
| 4. Acceptance | 5.14 | 1.65 |
| 5. Instrumental support | 5.12 | 1.74 |
| 6. Self-distraction | 5.03 | 1.75 |
| 7. Positive reframing | 4.97 | 1.68 |
| 8. Use of emotional support | 4.79 | 1.62 |
| 9. Venting items | 4.70 | 1.61 |
| 10. Humour | 4.52 | 1.75 |
| 11. Denial | 4.36 | 1.65 |
| 12. Self-Blame | 4.28 | 1.68 |
| 13. Behavioural disengagement | 3.99 | 1.49 |
| 14. Substance use | 3.47 | 1.63 |

Source: Field survey, (2021).

The results reveal that coping strategies significantly influenced health anxiety levels of health professionals ($F = 4.515$, $p<0.05$). The study demonstrated that coping strategies significantly accounted for 17.1% of variances in health anxiety among participants. Further analysis demonstrated the most significant coping strategy as predictor of health anxiety. The result of the analysis is presented in Table 5.

The results in Table 5 indicate that the coping strategies, which significantly contributed to health anxiety was Self-blame ($\beta = .132$, $p = .045$). This finding suggests that given the health anxiety of health professionals, self-blame plays a huge role.

## Coping strategies influence on stress levels of health professionals

To determine the influence of coping strategies on stress levels of participants. Multiple linear regression analysis was conducted, and the results of the analysis are presented in Table 6.

Results in Table 6 above reveal that coping strategies had a significant positive relationship with stress ($R = .478$, $p < 0.05$). and that coping strategies significantly influenced stress levels of participants. The findings indicate that coping strategies significantly accounted for 22.8% of variances in health anxiety among participants. Further analysis also revealed the significant coping strategy, which predicts stress, is presented in Table 7. The results indciacte that Active Coping as a significant determinant of stress among health professionals ($\beta = .253$, $p = .000$).

## Difference in health anxiety and stress level in terms of gender of health professional

Hypothesis three

**Table 4. Model summary of coping strategies influence on health anxiety of health professionals.**

| R | R Square | F Change | Sig. |
|---|---|---|---|
| .414 | .171 | 4.515 | .000 |

Dependent Variable: Health anxiety Significant at p<0.05.
Predictor: Coping Strategies.

**Table 5. Coping strategies as predictor of health anxiety of health professionals.**

| Model | Unstandardized Coefficients | | Standardized Coefficients | t | Sig. |
|---|---|---|---|---|---|
| | B | Std. Error | Beta | | |
| (Constant) | 27.576 | 1.803 | | 15.296 | .000 |
| Self-distraction | .245 | .286 | .062 | .859 | .391 |
| Active coping | -.198 | .272 | -.050 | -.729 | .466 |
| Denial | .374 | .262 | .090 | 1.429 | .154 |
| Substance use | .378 | .280 | .089 | 1.347 | .179 |
| Use of emotional support | -.423 | .305 | -.099 | -1.386 | .167 |
| Instrumental support | .504 | .285 | .127 | 1.770 | .078 |
| Behavioral disengagement | .547 | .310 | .119 | 1.764 | .079 |
| Venting items | .029 | .299 | .007 | .097 | .922 |
| Positive reframing | .586 | .303 | .143 | 1.934 | .051 |
| Planning | -.372 | .313 | -.086 | -1.188 | .236 |
| Humor | .071 | .250 | .018 | .282 | .778 |
| Acceptance | .108 | .300 | .026 | .362 | .718 |
| Religion | -.504 | .261 | -.126 | -1.927 | .055 |
| Self-blame | .540 | .268 | .132 | 2.013 | .045 |

Dependent Variable: Health Anxiety Significant p < 0.05.

$H_A3$: There is a significant difference among males and females' health professionals in terms of health anxiety and stress in selected hospitals in the Cape Coast Metropolis.

To determine whether a significant difference exists between male and female health professionals in terms of health anxiety and stress, an independent samples t-test was used. The result is presented in Tables 8 and 9.

The result from Table 8 shows that the mean score of male health professionals *(M = 35.14, SD = 8.26)* is significantly higher than that of female health professionals *(M = 34.95, SD = 6.42)*. The result further reveals that generally, a significant gender difference exists in the health anxiety of health professionals in the Cape Coast Metropolis *(t (320) = .210 p = .021)*. The results, therefore, suggest that male health professionals experienced significantly higher health anxiety than their female counterparts. Based on this finding, the alternative hypothesis is accepted against the null hypothesis.

According to the result of Table 9, there is no gender differences among male and female health professionals regarding experiences of stress *(t (320) = .301 p = 416)*. This is because the sig. value is greater than 0.05. Thus, the alternative hypothesis is rejected.

Hypothesis four

## Age categories difference in terms of coping strategies of health professionals

$H_A4$: There is a significant difference among the age category of health professionals in terms of coping strategies in selected hospitals in the Cape Coast Metropolis.

**Table 6. Coping strategies influence on stress levels of health professionals.**

| R | R Square | F Change | Sig. |
|---|---|---|---|
| .478 | .228 | 6.494 | .000 |

Dependent Variable: Stress LevelsSignificant at p<0.05.
Predictor: Coping Strategies.

**Table 7. Coping strategies as predictor of stress levels of health professionals.**

| Model | | Unstandardized Coefficients | | Standardized Coefficients | T | Sig. |
|---|---|---|---|---|---|---|
| | | B | Std. Error | Beta | | |
| | (Constant) | 19.186 | 1.388 | | 13.822 | .000 |
| | Self-distraction | .370 | .220 | .118 | 1.682 | .094 |
| | Active coping | .802 | .209 | .253 | 3.838 | .000 |
| | Denial | .269 | .202 | .081 | 1.334 | .183 |
| | Substance use | -.390 | .216 | -.115 | -1.805 | .072 |
| | Use of emotional support | -.166 | .235 | -.049 | -.708 | .480 |
| | Instrumental support | .110 | .219 | .035 | .501 | .617 |
| | Behavioural disengagement | .393 | .239 | .107 | 1.642 | .102 |
| | Venting items | .188 | .230 | .055 | .816 | .415 |
| | Positive reframing | -.196 | .233 | -.060 | -.843 | .400 |
| | Planning | .349 | .242 | .101 | 1.445 | .149 |
| | Humor | -.136 | .193 | -.043 | -.704 | .482 |
| | Acceptance | .372 | .231 | .112 | 1.612 | .108 |
| | Religion | -.099 | .202 | -.031 | -.490 | .625 |
| | Self-blame | -.095 | .206 | -.029 | -.462 | .644 |

Dependent Variable: Stress.

**Table 8. Gender difference among health professionals in terms of health anxiety.**

| Health Anxiety | | N | Mean | SD | *t* | *df* | *Sig.* |
|---|---|---|---|---|---|---|---|
| | Male | 80 | 35.14 | 8.26 | .210 | 320 | .021 |
| | Female | 242 | 34.95 | 6.42 | | | |

Significant, p< 0.05.

**Table 9. Gender difference among health professionals in terms of stress.**

| Stress | | N | Mean | SD | *t* | *df* | *Sig.* |
|---|---|---|---|---|---|---|---|
| | Male | 80 | 28.83 | 5.02 | .301 | 320 | .416 |
| | Female | 242 | 28.61 | 5.69 | | | |

Significant, p < 0.05.

Differences in coping strategies used by different age categories of health professionals were examined. The three levels of age categories were compared with coping strategies used. A one-way analysis of variances was used. Before this, preliminary analysis investigations, which support the use of the analytical tool, was conducted. They include Test of Normality and Homogeneity of Variances. The results of the preliminary analysis are presented in Tables 10 and 11.

According to the result of the normality test, the data is normally distributed. This is because the Sig. value of the Shapiro-Wilk Test for the age categories are greater than 0.05. A test of Homogeneity of Variances was conducted to confirm the assumption that justifies the use of the statistical tool *ANOVA*.

Table 10. Test of normality of age category and coping strategies.

| Age Category | Shapiro-Wilk | | |
|---|---|---|---|
| | Statistic | Df | Sig. |
| 18–29 years | .970 | 213 | .603 |
| 30–49 years | .954 | 107 | .474 |
| 50–60 years | .901 | 2 | .113 |

Significant p > 0.05.

Table 11. Test of homogeneity of age category and coping strategies.

| Levene Statistic | df1 | df2 | Sig. |
|---|---|---|---|
| 1.826 | 3 | 318 | .163 |

Significant p < 0.05.

Table 12. ANOVA age category of health professionals and coping strategies.

| Group | Sum of Squares | Df | Mean Square | F | Sig. |
|---|---|---|---|---|---|
| Between Groups | 922.157 | 3 | 461.078 | 2.340 | .098 |
| Within Groups | 62653.507 | 318 | 197.024 | | |
| Total | 14999.388 | 321 | | | |

Significant p < 0.05.

The results reveal that the sig. value is greater than 0.05 (p = .163) signifying that equal variances are assumed in the data. (See Table 12). Hence, an ANOVA analysis was computed to examine probable differences among health professionals with regard to experiences of health anxiety.

The results indicate that, in terms of coping strategies, there are no significant differences among the experiences of the three levels of age categories of health professionals [$F(3, 318) = 2.340$, $p = .098$].

## Discussions

The majority of participants, in terms of gender, were female health professionals This indicates that females are the majority in the health profession. Nurses dominated the study, representing approximately two-thirds of the sample. This result reflects the fact that nurses represent the highest percentage of health professionals in Ghana. This is similar to what previous studies have reported regarding females constituting the majority of the health workforce in the health profession. [48,49] The study found that health professionals adopt multiple coping strategies to mitigate the consequences of the COVID-19 pandemic on their well-being and performance at the workplace. Significant coping strategies include active coping, planning, religion, acceptance, instrumental support, humour, and seeking emotional support. Active coping indicates that most participants usually become conscious of the impact of the pandemic in their life and devise appropriate mechanisms to eliminate its devastating impact, including positive reframing, venting items, behavioural disengagement, and self-distraction. Similarly, most participants adopted religious practices as a laudable means of coping. This reflects a faith-based attitude or practice that shields individuals from the possible impact of

the pandemic. Many people utilize religion as a method of mental or physical improvement as religiosity has been linked to higher levels of mental and physical health when incorporated into treatment plans [50,51]. The potential for religion as a coping strategy has long been discussed [53] According to Pargament [52] religious coping, while generally related to ideas about the sacred, encompasses a wide range of efforts informed by one's religion to cope with life stressors.

These include behaviours such as prayer, confession, seeking spiritual support from clergy or others, and acceptance of circumstances as representing the will of God. Religious coping may be beneficial or costly depending on whether one engages in positive or negative coping methods [53] Other significant responses include *Acceptance*, this connotes the idea that participants acknowledge that the pandemic is real and adopt appropriate safety precautions. It helps to prevent the pain related to the pandemic from worsening into more suffering. This strategy, *Instrumental Support*, adopted by participants is because during the current pandemic, health professionals receive tangible help and support from other people, and this has become means of coping because their efforts and role in the pandemic have been acknowledged by others. The incentives and tax-free allowances offered by the government of Ghana to healthcare professionals are forms of instrumental support. The result with emphasis on multiple coping styles among healthcare providers was accentuated by Munawar and Choudhry [54] According to their study, frontline workers adopted coping strategies such as reducing media contact, having low inquiry into COVID-19 details, and using religious coping, among others. Planning reflects the fact that participants create specific tactics to overcome anticipated stressors and anxieties related to the pandemic that may affect their wellbeing and work performances. Similarly, most participants adopted religious practices as a laudable means of coping. This reflects a faith-based attitude or practice that shields individuals from the impact of the pandemic.

The results of the study are consistent with the work by Ofori *et al [8]*. (and Asare-Nuamah, Onumah, Dick-Sagoe, and Kessie [9] that praying more often, positive attitude from colleagues, the government's tax-free relief, salary, and drinking locally prepared herbs were coping mechanisms during Covid -19 pandemic. Similarly, Shechter *et al.* [55]) found that the most dominant coping strategies adopted by healthcare practitioners at the time of the study were physical exercises, contacting counselling by therapists and online counselling guide (social support).

Furthermore, the results showed that coping strategies had a positive connection with health anxiety and stress. Further analyses showed that coping strategies accounted for variances in health anxiety and stress among health professionals. These results showed that coping strategies have significant impact on experiences of health anxiety and stress. Thus, using the right coping strategies can either improve the wellbeing of healthcare professional experiencing psychological distresses associated with the pandemic or may woefully compound the already affected psychological and emotional wellbeing.

A study by Fullana *et al.* [56] (which is consistent with the result of this study, identified that adherence to simple coping behaviours improves adjustment to symptoms of anxiety and depression. Similarly, Littleton, Horsley, John, and Nelson [57] found that coping strategies had an impact on psychological distress. Some empirical research [58,59]) have also found that coping strategies adopted during stressful situation impact the psychological and overall wellbeing of population. Invariably it means participants usually become aware of the stressor and anxiety and make conscious efforts to reduce the negative consequences of the stress. In essence health professionals can recognize the sources of stressors, and devise strategies to overcome the effects. The empirical evidence presented support the findings of the current

study that coping strategies have a significant relationship on psychological distress and, eventually, impact the level of experiences of psychological distress.

It was found that male health professionals experience significantly higher health anxiety than female health professionals. Conversely, no significant gender disparity among health professionals, in terms of stress, was identified. Furthermore, Mirzabeigi, Agha Mohammad Hasani, Sayadi, Safarian, and Parand Afshar [60] support the results of our currents study. Their study assessed health anxiety among healthcare personnel during the current pandemic. The study discovered that male healthcare providers expressed higher health anxiety than their female counterparts. Contrary to our current findings, AlAteeq, Aljhani, Althiyabi, and Majzoub) [61] discovered that males were significantly less likely to have anxiety.

The findings show that demographic features, such as gender, are determining components in experiences of psychological distress. This is consistent with Islam *et al*. [62] who examined panic disorder and generalized anxiety associated with COVID-19 among Bangladesh's general population.

The results of the study showed that in terms of coping strategies, there is no significant difference in the experiences of the three levels of age categories of health professionals [*F(3, 318) = 2.340*, *p = .098*]. It could be deduced from the results that health professionals did not differ on coping strategies adopted in the face of health anxieties and stress during the current pandemic. Since most of the participants were between "*18–29 years*" (n = 213, 66.2%,), one expects a difference in coping because younger health professionals have a high risk of stress and health anxiety, since they have less experience in the workplace. It appears that, during the pandemic, health professionals' experiences of stress may range from an acute stress which lasts briefly and that may not require less coping strategies [63] to participants having a coping positive appraisal strategy which is a type of coping strategy where an individual reframes a situation and tries only to see the positive side of the event at hand [64] Furthermore, it may appear that participants, irrespective of age differences, have no misconceptions of having the disease of health anxiety, or fear of the disease, or incapability to cope with the disease and the inadequacy of medical treatments [65].

## Limitation

First, all variables under consideration were assessed with self-reported measures, which may result in single-source bias. The descriptive design only describes what happens at the time of data collection; therefore, findings cannot be generalized to other areas in the country at a different time interval. Another limitation of the study was that during the data collection, the estimated number of medical doctors for the study had not been reached. Out of the proposed 67 medical doctors, only 19 willingly participated in the study, representing a returning rate of 28.4%. In this regard, generalization of the findings of the study is very limited with regard to medical doctors.

## Conclusions

Health professionals adopted diverse coping strategies ranging from positive to negative coping styles to overcome the negative impact of the COVID-19 pandemic. While some are relevant for positive change, others, such as *Self-distraction*, *Denial*, *Substance use*, *Behavioural disengagement*, and *Self-Blame* have the potential to cause further psychological and physiological disorders, thus harming the wellbeing of health professionals. It could be asserted that health professionals adopted mechanisms to fight the psychological disturbances imposed by the results of the pandemic. For instance, the study indicated that *Active Coping* was a significant predictor of variances in stress among health professionals. Similarly, *self-blame* and

*positive reframing* accounted for changes in health anxiety. Judging from the predictive role of coping strategies in health anxiety and stress, it is concluded that health professionals who adopt appropriate means of coping, such as *active*, *instrumental support*, *emotional support*, *acceptance*, and *humour* would be much successful in overcoming the psychological consequences of the pandemic. The study found that Coping strategies significantly influenced health anxiety ~~the~~ levels of health professionals, and that Active Coping is a significant determinant of stress among health professionals.

Age of health professionals does not determine the type of coping strategy they adopted during the pandemic. To ensure that health professionals execute their duties appropriately during this period, employers and stakeholders may be obliged to provide adequate support. The findings would also identify the coping strategies adopted by health professionals during experiences of psychological distresses due to COVID-19. Armed with this knowledge, appropriate support that meets the psychological or mental health needs of health workers can be designed in line with their coping strategies.

## Implication for clinical practice

The results of the current study have implications for health practitioners amid any covid pandemic. The findings showed that most participants were female health professionals. The~~is~~ results suggest that females constitute the majority in the health workforce in the Cape Coast Metropolis; therefore, research work in coping strategies of health workers that excludes female majority may be biased.

The findings show, also, that nurses dominated the study. They represented approximately two-thirds of the sample, implying that nurses are in the majority when it comes to health care professionals and, therefore, any measure of health anxiety and coping strategies used would impact more health care professional if nurses are considered first.

The current study found that health professionals adopt multiple coping strategies to mitigate the consequences of COVID-19 pandemic on their well-being and performance at the workplace. This implies that in the future a good solution to handling health anxiety among health professionals would be to employ multiple coping strategies to address a psychological health crisis. For instance, the tool of Active Coping strategies, such as an awareness of the stressor, followed by attempts to reduce the negative outcome, are preferred for use by healthcare professionals to employing an avoidant style. Secondly, the use of planning implies that healthcare professionals prefer to anticipate difficulties, barriers and anxieties related to the COVID-19 pandemic that that may affect their wellbeing and work performances to ~~rather than~~ avoiding performing their duties.

The findings also showed that most participants utilized religion as a method of coping. This implies that health professionals acknowledge that religion is good enough to be depended on in times of world psychological trauma. This means that health professionals preferred to pray, confess their faith in their God and chose spiritual support by the clergy or others in a time of a pandemic.

The study found that health professionals adopted Acceptance as a coping strategy. This implies that healthcare professionals were not in denial that the COVID-19 was real and, therefore, adopted relatively appropriate safety precautions.

Participants also welcomed Instrumental Support as a way of coping. This implies that during pandemics tangible support from government, such as tax reliefs and incentives, may be welcome to help healthcare workers~~,~~ cope. This does not exclude tangible and technical support from philanthropists, donor partners or any quarters available.

The findings also show that coping strategies have a significant impact on experiences of health anxiety and stress. This implies that coping strategies influence whether healthcare professionals will experience health anxiety or not, and the extent to which it may occur. This means that using the appropriate coping strategies can improve the wellbeing of healthcare professionals experiencing psychological distress.

It was also found that male health professionals experienced significantly higher health anxiety than female counterparts. This implies that gender of health professionals is a determining component during psychological distress; therefore, demographic features on gender and how they relate to coping should not be ignored during periods of such distress.

## Recommendations

1. Healthcare professionals should periodically assess the coping strategies adopted to overcome the psychological impacts related to their work during the pandemic. Such an assessment should go side by side with the inoculation of healthcare workers against the devastating impact of a pandemic. This will ensure adopting the right coping skills to deal with such instances.

2. Consistent screening of medical personnel, especially those directly involved in diagnosing and treating patients with COVID-19, should be done for evaluating indicators of psychological symptoms such as stress, depression, and anxiety, by using multidisciplinary mental health teams, including counsellors, psychiatrists, psychologists, and medical officers.

3. The counsellors and psychologists who provide interventions for health professionals must tailor their proceedings to be gender and age specific since these characteristics of healthcare professionals have a bearing on their psychological distress.

## Supporting information

**S1 Data. Data save.**
(DOCX)

## Author Contributions

**Conceptualization:** Bridgette Baaba.

**Data curation:** Anthony K. Nkyi, Bridgette Baaba.

**Formal analysis:** Anthony K. Nkyi, Bridgette Baaba.

**Investigation:** Anthony K. Nkyi, Bridgette Baaba.

**Methodology:** Anthony K. Nkyi, Bridgette Baaba.

**Project administration:** Anthony K. Nkyi, Bridgette Baaba.

**Writing – original draft:** Anthony K. Nkyi, Bridgette Baaba.

**Writing – review & editing:** Anthony K. Nkyi.

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
