## [Decision Letter · Decision Letter 0]

2 Aug 2023

PONE-D-23-14016COPING STRATEGIES AS PREDICTORS OF HEALTH ANXIETY AND STRESS AMONG HEALTH PROFESSIONALS DURING COVID-19 PANDEMIC IN CAPE COAST METROPOLIS, GHANA.PLOS ONE

Dear Dr. NKYI,

Thank you for submitting your manuscript to PLOS ONE. After careful consideration, we feel that it has merit but does not fully meet PLOS ONE’s publication criteria as it currently stands. Therefore, we invite you to submit a revised version of the manuscript that addresses the points raised during the review process.

We look forward to receiving your revised manuscript.

Kind regards,

Carmen Concerto

Academic Editor

PLOS ONE

Reviewers' comments:

Reviewer's Responses to Questions

**Comments to the Author**

1. Is the manuscript technically sound, and do the data support the conclusions?

Reviewer #1: Partly

Reviewer #2: Partly

2. Has the statistical analysis been performed appropriately and rigorously? 

Reviewer #1: Yes

Reviewer #2: No

3. Have the authors made all data underlying the findings in their manuscript fully available?

Reviewer #1: Yes

Reviewer #2: Yes

4. Is the manuscript presented in an intelligible fashion and written in standard English?

Reviewer #1: No

Reviewer #2: No

5. Review Comments to the Author

Reviewer #1: Dear Authors,

Upon conducting a thorough review of your manuscript, a number of critical aspects warrant further attention and revision:

1. Ethical Considerations: Your manuscript currently lacks sufficient detail in relation to how participant confidentiality was assured.

2. Data Accessibility: More precise information about how and where the data can be accessed would greatly enhance the transparency of your study.

3. Recruitment: The clarity of your procedure is somewhat lacking. While it appears that you have employed the optimal Multistage Sampling technique, the recruitment procedures outlined are presented in a vague and confusing manner. Furthermore, the absence of a comprehensive demographic report hinders our ability to determine the congruence of the sampling approach.

4. Scales used: The utilization of two stress scales warrants proper justification in order to enhance clarity and academic rigor. It is imperative to provide a more thorough explanation regarding the rationale behind the chosen approach. It is important to bear in mind that the subsequent discussion should extensively analyze and highlight the disparities stemming from the utilization of these two stress scales.

5. Results: There seems to be an omission of results concerning gender differences analyses, despite it being one of your main hypotheses. Furthermore, the data presented in Table 3 raises concerns. Despite text suggesting that positive reframing (β = .143, p = .051) significantly contributes to health anxiety, the p-value reported exceeds the conventionally accepted threshold of significance (typically set at .05 or lower). In addition, it is worth noting that data interpretation occasionally appeared to be interpolated in the presented findings, as it should ideally be situated within the discussion section.

6. Comprehensive Reporting: For the benefit of a thorough interpretation of your study, ensure all data analyses and relevant demographic information are included in your report in a detailed manner. For instance, demographic data differentiating nurses from other health professionals in each facility, along with details on work experience / age for these groups, should be included to offer a more complete understanding of your findings. By presenting all demographics and relevant analyses, you can avoid selective reporting, thereby enhancing the validity and robustness of your results.

7. Discussion: The clarity and coherence of your discussion section could be improved. The implications and significance of your hypothesis need to be more explicitly addressed, as the current explanation of the regression data is unclear. In addition, reference to prior studies that provide a broader context to your findings could be beneficial. It is evident that some of the referenced citations do not adequately support your hypothesis or clarify the points you are making. You should also explicitly state the correlational nature of the study design and the limitations inherent in establishing causality. For example, considering potential bidirectional relationships and unaccounted variables that may influence the observed associations, such as the potential confounding factor of the female gender and nursing profession, could enhance your discussion. You have sometimes made unsupported claim based on your data in the discussion, and that should be revised.

8. Study Limitations: An explicit discussion addressing the limitations of your study, such as potential recruitment biases or issues of generalizability, is recommended. Incorporating this information will contribute to strengthening your paper's reliability.

9. Recommendations: Strengthen your recommendations section by suggesting more specific strategies for implementing your findings in clinical practice. This will make your recommendations more useful and applicable.

10. Conclusion: Your conclusion would benefit from more explicit suggestions for future research, derived from your own findings. This will provide clearer guidance for future investigations in this field. Avoid also here unsubstantiated claims (see point 7).

E.g. The interpretation of the data you proposed is incongruent with the obtained results: the absence of correlation does not necessarily indicate a positive impact.

11. Readability and Language Quality: I recommend a comprehensive proofreading to enhance readability and rectify any English language errors. The current state of the text exhibits a certain degree of compromised fluency, thereby necessitating and affording an opportunity for improvement.

In summary, your study has notable potential, due to the subject matter at hand and the type of data gathered, but it requires addressing these issues through major revisions before it is suitable for publication.

Your main concern should be to ensure comprehensive reporting of all results, including those pertaining to gender differences, which appear to be omitted currently. Present in a better way demographic data and strive to conduct thorough analyses on any data that may introduce confounding variables and potentially influence the outcomes of the study. I highly recommend thoroughly revising the results section in its entirety due to the possibility of reported errors, such as the one mentioned earlier.(point 5)

It is very important to provide a more detailed discussion of your findings, as your manuscript currently lacks a comprehensive framework for interpretation. For example, what might be the implications of the observed positive linear regression between specific coping strategies and stress levels? Could this suggest inefficacy, or is there another potential explanation? Of course, the aforementioned questions provided are merely a limited selection intended solely for illustrative purposes; you should give the reader a way to navigate throughout your data; providing a more detailed interpretation will give your readers a clearer understanding of your results and their significance. Exercise caution when discussing the implications of your findings; remember that correlation does not imply causation and that lack of significance does not mean lack of an effect. You have also to elaborate thoughtfully on the potential implications of your research for both academic investigations and clinical practice.

Finally, please kindly review and amend the written format and the organization of the sections, as they do not conform to best conventions.

Reviewer #2: Dear Authors,

I have recently reviewed your manuscript titled "COPING STRATEGIES AS PREDICTORS OF HEALTH ANXIETY AND STRESS AMONG HEALTH PROFESSIONALS DURING COVID-19 PANDEMIC IN CAPE COAST METROPOLIS, GHANA" which aims to assess the prevalence of coping strategies, stress levels, and hypochondria in a population of Ghanaian healthcare professionals involved in managing the COVID-19 pandemic. This is an important area of study, and your efforts to contribute to this body of knowledge are commendable.

However, after a careful review, I have identified several areas that need significant improvement before the manuscript can be considered for publication.

Firstly, while the topic of your study is of interest, the manuscript does not seem to add significant new information to the existing literature. The positive association between the implementation of coping strategies and levels of stress and hypochondria is already well-documented.

Secondly, the manuscript contains numerous formatting errors that make it difficult to follow. The language used needs to be improved to meet the standard expected for a scientific paper. There are instances of unclear or awkward phrasing that detract from the overall readability of the text. I would strongly recommend a thorough proofreading and editing of the manuscript to improve its clarity and readability.

Furthermore, there are missing data points in your manuscript, which further hinder the comprehension of the study's findings. Please ensure that all necessary data are included and clearly presented.

In light of these issues, I recommend a thorough revision of the manuscript. Addressing these concerns could significantly improve the quality of your work and its suitability for publication.

Thank you for your contribution to this important field of study. I look forward to seeing a revised version of your manuscript.

Best regards,

Antonio Di Francesco

6. PLOS authors have the option to publish the peer review history of their article (what does this mean?). If published, this will include your full peer review and any attached files.

Reviewer #1: **Yes: **Pierfelice Cutrufelli

Reviewer #2: **Yes: **Antonio Di Francesco

---

## [Author Response · Author response to Decision Letter 0]

10 Oct 2023

Dear Pierfelice Cutrufelli 

Reviewer #1: 

1. Ethical Considerations: Your manuscript currently lacks sufficient detail in relation to how participant confidentiality was assured. 

Response: Participant confidentiality assured revised. 

2. Data Accessibility: More precise information about how and where the data can be accessed would greatly enhance the transparency of your study.

Response: Data accessibility included.

3. Recruitment: The clarity of your procedure is somewhat lacking. While it appears that you have employed the optimal Multistage Sampling technique, the recruitment procedures outlined are presented in a vague and confusing manner. Furthermore, the absence of a comprehensive demographic report hinders our ability to determine the congruence of the sampling approach.

Response: Demographic report included. Table 1 

4. Scales used: The utilization of two stress scales warrants proper justification to enhance clarity and academic rigor. It is imperative to provide a more thorough explanation regarding the rationale behind the chosen approach. It is important to bear in mind that the subsequent discussion should extensively analyze and highlight the disparities stemming from the utilization of these two stress scales.

Response: Justification made and discussion made 

5. Results: There seems to be an omission of results concerning gender differences analyses, despite it being one of your main hypotheses. Furthermore, the data presented in Table 3 raises concerns. Despite text suggesting that positive reframing (β = .143, p = .051) significantly contributes to health anxiety, the p-value reported exceeds the conventionally accepted threshold of significance (typically set at .05 or lower). In addition, it is worth noting that data interpretation occasionally appeared to be interpolated in the presented findings, as it should ideally be situated within the discussion section.

Response: Omission included and data in Table 3 Corrected. Positive reframing and other errors corrected. 

6. Comprehensive Reporting: For the benefit of a thorough interpretation of your study, ensure all data analyses and relevant demographic information are included in your report in a detailed manner. For instance, demographic data differentiating nurses from other health professionals in each facility, along with details on work experience / age for these groups, should be included to offer a more complete understanding of your findings. By presenting all demographics and relevant analyses, you can avoid selective reporting, thereby enhancing the validity and robustness of your results.

Response: relevant demographic information included 

7. Discussion: The clarity and coherence of your discussion section could be improved. The implications and significance of your hypothesis need to be more explicitly addressed, as the current explanation of the regression data is unclear. In addition, reference to prior studies that provide a broader context to your findings could be beneficial. It is evident that some of the referenced citations do not adequately support your hypothesis or clarify the points you are making. You should also explicitly state the correlational nature of the study design and the limitations inherent in establishing causality. For example, considering potential bidirectional relationships and unaccounted variables that may influence the observed associations, such as the potential confounding factor of the female gender and nursing profession, could enhance your discussion. You have sometimes made unsupported claims based on your data in the discussion, and that should be revised.

Response: corrections effected and other errors deleted . 

8. Study Limitations: An explicit discussion addressing the limitations of your study, such as potential recruitment biases or issues of generalizability, is recommended. Incorporating this information will contribute to strengthening your paper's reliability.

Reponses: Limitations of the study addressed. 

9. Recommendations: Strengthen your recommendations section by suggesting more specific strategies for implementing your findings in clinical practice. This will make your recommendations more useful and applicable.

Response: Recommendations strengthened.

10. Conclusion: Your conclusion would benefit from more explicit suggestions for future research, derived from your own findings. This will provide clearer guidance for future investigations in this field. Avoid also here unsubstantiated claims (see point 7).

E.g. The interpretation of the data you proposed is incongruent with the obtained results: the absence of correlation does not necessarily indicate a positive impact.

Response: Corrections effected. 

11. Readability and Language Quality: I recommend a comprehensive proofreading to enhance readability and rectify any English language errors. The current state of the text exhibits a certain degree of compromised fluency, thereby necessitating and affording an opportunity for improvement.

Response: Comprehensive proofreading made 

Dear Antonio Di Francesco

Reviewer #2: 

 COPING STRATEGIES AS PREDICTORS OF HEALTH ANXIETY AND STRESS AMONG HEALTH PROFESSIONALS DURING COVID-19 PANDEMIC IN CAPE COAST METROPOLIS, GHANA

Firstly, while the topic of your study is of interest, the manuscript does not seem to add significant new information to the existing literature. The positive association between the implementation of coping strategies and levels of stress and hypochondria is already well-documented.

Response: Significant information added to the literature. Topic of the paper amended due to some errors made in the analysis. 

Secondly, the manuscript contains numerous formatting errors that make it difficult to follow. The language used needs to be improved to meet the standard expected for a scientific paper. There are instances of unclear or awkward phrasing that detract from the overall readability of the text. I would strongly recommend a thorough proofreading and editing of the manuscript to improve its clarity and readability.

response: comprehensive review and proofreading made 

Furthermore, there are missing data points in your manuscript, which further hinder the comprehension of the study's findings. Please ensure that all necessary data are included and clearly presented.

Response: Addressed 

In light of these issues, I recommend a thorough revision of the manuscript. Addressing these concerns could significantly improve the quality of your work and its suitability for publication.

Response: Concerns Addressed

---

## [Editor Report · Decision Letter 1]

9 Nov 2023

PONE-D-23-14016R1COPING, HEALTH ANXIETY, STRESS AMONG HEALTH PROFESSIONALS DURING COVID-19, CAPE COAST, GHANA.PLOS ONE

Dear Dr. NKYI,

Thank you for submitting your manuscript to PLOS ONE. After careful consideration, we feel that it has merit but does not fully meet PLOS ONE’s publication criteria as it currently stands. Therefore, we invite you to submit a revised version of the manuscript that addresses the points raised during the review process.

ACADEMIC EDITOR: The discussion could benefit from a more thorough exploration of the data's implications and relevance to the field. This would not only respond to previous feedback but also enhance reader engagement with your findings.While improvements have been made in the language and demographic details, there is still room for refinement to meet publication standards. It might be beneficial to consider additional language editing services to assist with this.

We look forward to receiving your revised manuscript.

Kind regards,

Carmen Concerto

Academic Editor

PLOS ONE

Journal Requirements:

Additional Editor Comments:

Dear Authors,

the discussion could benefit from a more thorough exploration of the data's implications and relevance to the field. This would not only respond to previous feedback but also enhance reader engagement with your findings.

While improvements have been made in the language and demographic details, there is still room for refinement to meet publication standards. It might be beneficial to consider additional language editing services to assist with this.

---

## [Author Response · Author response to Decision Letter 1]

5 Dec 2023

Response to the academic editor,

Thank you for your efforts to improve my manuscript to meet PLOS ONE standards. Being guided by the comments made by the academic editor, I have improved the data implication and its relevance to the field. I have also engaged additional language editing services to help me improve on the manuscript. I will be grateful if my manuscript fits the standard for approval and publication.

---

## [Editor Report · Decision Letter 2]

18 Dec 2023

COPING, HEALTH ANXIETY, STRESS AMONG HEALTH PROFESSIONALS DURING COVID-19, CAPE COAST, GHANA.

PONE-D-23-14016R2

Dear Dr. NKYI,

We’re pleased to inform you that your manuscript has been judged scientifically suitable for publication and will be formally accepted for publication once it meets all outstanding technical requirements.

Kind regards,

Carmen Concerto

Academic Editor

PLOS ONE
---

## [Editor Report · Acceptance letter]

17 Jan 2024

PONE-D-23-14016R2 

PLOS ONE

Dear Dr. Nkyi, 

I'm pleased to inform you that your manuscript has been deemed suitable for publication in PLOS ONE. Congratulations! Your manuscript is now being handed over to our production team.

Kind regards, 

on behalf of

Dr. Carmen Concerto 

Academic Editor

PLOS ONE